# Responses to concerns about child maltreatment: a qualitative study of GPs in England

-1

Jenny Woodman,[1] Ruth Gilbert,[1] Janice Allister,[2] Danya Glaser,[3] Marian Brandon[4]

[1]Department of Paediatric Epidemiology and Biostatistics, UCL-Institute of Child Health, London, UK
[2]Royal College of General Practitioners, London, UK
[3]Great Ormond Street Hospital for Children NHS Foundation Trust London, London, UK
[4]Centre for Research on Children and Families, Elizabeth Fry Building, University of East Anglia, Norwich, UK

Correspondence to
Jenny Woodman;
j.woodman@ucl.ac.uk,
jenny.woodman@gmail.com

## ABSTRACT

**Objectives:** To provide a rich description of current responses to concerns related to child maltreatment among a sample of English general practitioners (GPs).

**Design:** In-depth, face-to-face interviews (November 2010 to September 2011). Participants selected and discussed families who had prompted 'maltreatment-related concerns'. Thematic analysis of data.

**Setting:** 4 general practices in England.

**Participants:** 14 GPs, 2 practice nurses and 2 health visitors from practices with at least 1 'expert' GP (expertise in child safeguarding/protection).

**Results:** The concerns about neglect and emotional abuse dominated the interviews. GPs described intense and long-term involvement with families with multiple social and medical problems. Narratives were distilled into seven possible actions that GPs took in response to maltreatment-related concerns. These were orientated towards whole families (monitoring and advocating), the parents (coaching) and children (opportune healthcare), and included referral to or working with other services and recording concerns. Facilitators of the seven actions were: trusting relationships between GPs and parents, good working relationships with health visitors and framing the problem/response as 'medical'. Narratives indicated significant time and energy spent building facilitating relationships with parents with the aim of improving the child's well-being.

**Conclusions:** These GPs used core general practice skills for on-going management of families who prompted concerns about neglect and emotional abuse. Policy and research focus should be broadened to include strategies for direct intervention and on-going involvement by GPs, such as using their core skills during consultations and practice systems for monitoring families and encouraging presentation to general practice. Exemplars of current practice, such as those identified in our study, should be evaluated for feasibility and acceptability in representative general practice settings as well as tested for efficacy, safety and cost. The seven actions could form the basis for the 'lead professional' role in general practice as proposed in the 2013 version of 'Working Together' guidance.

## Strengths and limitations of this study

- This study generated hypotheses about responses that were feasible in English practices with some expertise and interest.
- Participant accounts were detailed and candid, and findings resonate with other research in general practice settings.
- Owing to a small and non-random sample, results cannot be generalised to all general practices in England. Although our results confirm those from other studies, it would be helpful for a similar study to be undertaken with a different sample in order to identify any additional responses.

in England each year.[1 2] All healthcare professionals have a statutory duty to protect children from child maltreatment.[3] General practitioners (GPs) are uniquely placed to respond because they offer services to the whole family often over many years, manage parental problems that put children at risk of child maltreatment, such as mental health and substance misuse,[4] and are skilled in fostering relationships, which constitute an important element of social welfare interventions. Although identification could undoubtedly be improved, GPs in England already record maltreatment-related problems in at least 1% of all children registered with them.[5] The true figure for children who raise concerns for GPs is likely to be far higher.[6]

Many children who have their maltreatment-related problems identified will not meet the high thresholds for action by children's social care, which result from social workers prioritising scarce resources in an overstretched service.[2 7–10] Academics are increasingly recognising that professionals require a range of responses for maltreatment-related concerns, including but not limited to referral to and joint working with children's social care.[7] This appears to be reflected in policy and good practice guidelines for GPs which recommend that GPs record and monitor concerns, gather

## INTRODUCTION

Child maltreatment (abuse or neglect) is common, affecting at least 4% of all children

information, discuss with colleagues, hold team meetings and, where thresholds are met, refer the family to children's social care.[11–13] However, a closer look at these documents reveals that these recommendations focus on improving recognition of maltreatment, helping health professionals to make decisions about when it is appropriate to refer a child to children's social care and contributing to social care processes. An exception is new (2013) statutory guidance which provides a description of direct intervention by GPs for some children below the threshold for children's social care intervention. This 'lead' role is described as supporting the family, acting as an advocate and coordinating support services.[3] There is a lack of other detail about the suggested responses and it is unclear how they might be put into practice or what skills, resources or service context would be needed.

Similarly, there is a lack of empirical research in this area. The existing research tends to conceptualise 'response' as referral to children's social care[14] and focuses exclusively on GP participation in social care processes[15 16] or identification of maltreatment-related problems.[17–19] One exception is a large mixed methods study by Tompsett et al[20] which aimed to explore the nature and consequences of conflicts of interests for English GPs in safeguarding children, though the scope of the findings were much broader than its original aim suggests. The study consisted of: a literature review; a survey of 96 English GPs, in-depth interviews with GPs (N=14); interviews with key stakeholders (N=19); three focus groups with young people, young mothers and a minority ethnic group and a Delphi consensus about the guiding principles of GPs in safeguarding children (with 25 experts). Data were collected between 2006 and 2007. To our knowledge, this study is the only existing source of empirical data about how GPs are responding to concerns about maltreatment in an English setting. The study identified four roles that GPs played in responding to maltreated children and reported exemplars of good practice for GPs.

We aimed to contribute to the scant research literature on how GPs in England can respond to maltreatment-related concerns by conducting an in-depth qualitative study asking how a small sample of GPs understood and responded to child maltreatment-related concerns in their daily practice.

## METHODS

One researcher conducted in-depth individual interviews with 14 GPs, two practice nurses and two health visitors from four GP practices in England. This article focuses largely on data from the GP interviews. The practices were known to the research team via a previous research study.[6] The four practices were chosen to include geographical spread across England, to have child protection expertise (at least one 'expert' GP who was a named doctor for child protection (1 GP), had delivered child protection training (all 4 GPs) or had

contributed to relevant policy (3 GPs)). All four practices had regular discussion of child protection concerns at clinical meetings and two of the four practices had health visitors based on site. The practices had between three and six full-time-equivalent GPs. At three of the practices four GPs were interviewed and at the remaining practice four GPs were interviewed. Participants at each practice were recruited through the gatekeeper 'expert' GP and researcher visits to the practices. The research team met and corresponded with the four gatekeeper GPs during the study set-up and recruitment phase. These gatekeeper GPs were also interviewed. Two pilot interviews were conducted.

By establishing trust and rapport with the participant in individual interviews, we hoped to elicit 'private' account of experiences, attitudes and beliefs in order to understand what *happened* in primary care.[21–23] 'Private' accounts have been defined as those which tend to contain more controversial views and be based on real experiences, with all their complexity and difficulty.[24] 'Public' accounts, on the other hand, tend to confirm the dominant ideology (in our case, what GPs think they *should* be doing).[24] Asking participants to recount narratives based on experience also helps to elicit accounts that move beyond the socially acceptable or familiar.[22] A study using focus groups to investigate child safeguarding by GPs in Denmark noted that the GPs appeared to be most comfortable with case-based discussion[17] and this approach appeared to be acceptable to participants and to generate rich data in our two pilot interviews.

In the interviews, the researcher elicited narratives by asking the participants to choose two or three 'children, young people or families who had prompted maltreatment-related concerns' and describe their concerns and involvement. In keeping with the aim of allowing participants to tell their stories and control the content, the interviews were free ranging with minimal steering from the researcher. Similarly, we did not specify whether the participants should choose children already known to or working with children's social care or whether the concern should be current or historical.

Our study design allowed for families to be discussed by two or more participants from the same practice and each expert GP spoke to colleagues to clarify whether this had been the case. However, the number of cases in which this occurred (only two families were discussed by more than one GP) was small and not commented on further in this article. Interviews were face to face, conducted between November 2010 and September 2011, lasted an average of 50 min and were audio recorded and later transcribed. In total, we collected 837 min of interview data from 17 participants (602 min from the 14 GP participants).

We used thematic analysis with an inductive and interpretive approach.[22 25] The exception to this was our a priori interest in whether and how GPs recorded concerns to inform our population-based analyses measuring GP practice.[5] Using NVivo software, one researcher

systematically assigned to each segment of interview transcript one or more concept labels (open coding). She made constant comparisons of codes within and between interviews to generate more abstract themes and build-up an understanding of the relationships between them. The abstract themes and understanding of relationships between them were refined by paying particular attention to data that did not fit and using reflections on these instances. We sought participant views on our preliminary results via an e-leaflet. Seven participants (5 GPs) responded, including at least one from each data collection site. This feedback was incorporated into the final interpretation. One researcher (the interviewer) conducted the coding and analysis with support from a senior researcher who independently coded two transcripts. The wider research team probed and questioned interpretation throughout the study.

This study was conducted as part of a PhD award and more detailed results can be found in the first author's thesis, due to be published in 2014.

## RESULTS

The GP participants tended to be experienced professionals (average 19 years since qualification; range 5–40 years) who had worked for long periods within their current practice (average 10 years; range 6 months to 23 years). The GPs discussed 26 different families (range 1–3 families per participant).

The data generated themes which we grouped as answers to three overarching questions: To *whom* were the GPs responding and *why these families*? What *actions* did they describe taking? What were the important *facilitators or barriers* for these actions? These questions were identified during data analysis.

### To whom

The GP narratives about families were coded as four broad types, which we named using quotes from the interviews:

1. 'stable at this point in time but it's a never ending story': narratives describing families with previous very serious child protection concerns who had since achieved a fragile stability that participants perceived to require extra vigilance on their behalf. The current concerns were about neglect and emotional abuse.
2. 'on the edge': narratives describing families who were barely coping and perceived as liable to tip over the edge at any moment. Concerns were about neglect and, to a lesser degree, emotional abuse.
3. 'was it, wasn't it': narratives describing situations where participants had a high degree of uncertainty as to whether physical or sexual abuse had taken place and where much time was spent trying to establish whether the abuse was likely to have occurred.
4. 'fairly straightforward': uniformly brief narratives in which there was high certainty about physical or sexual abuse and decisive onwards referrals.

In some cases, it was clear how the participants' views of the family had evolved over time and, for this reason, some of the 26 families were classified as more than one family type (see table 1). 'Stable at this point' and 'on the edge' families were discussed with the highest frequency (see table 1) and occupied most talk time. For these families, the participants could give a high level of detail about multiple family members, often reaching back many years. These two family types prompted concerns about neglect and emotional abuse and it was these concerns that dominated the interviews.

> Neglect really. I think with chaotic lifestyles that the child may become… well just not be cared for adequately. […] Parents who become impoverished because of their drug using behaviour are at just that much more risk of physical neglect of not feeding the child, not caring for the child, not changing its nappy, of not…and to an extent emotional neglect as well, just that there's not enough parenting input.

(Participant 14; 7-month-old baby)

> I'm not worried about the children whether they will be abused physically, I'm worried about the emotional deprivation rather than… the neglect rather than the abuse.

(Participant 15, two children aged 9 and 11 years)

For 'on the edge' and 'stable at this point' families, parental behaviour was commonly described in terms of 'low parenting capacity', 'poor parenting' or 'impoverished' parenting. The participants recounted how they were concerned that these parents failed to supervise their children adequately, transferred parenting responsibilities onto older siblings who were themselves young children, failed to set boundaries, routines or bedtimes, allowed children to miss school, did not adequately comply with essential medical care for their children and, in some cases, might not be able to keep young children clean and fed.

Although we did not systematically collect information on the current status of each case with children's social care, the contact between this agency and the families was mentioned in many interviews. 'On the edge' and 'stable at this point' families were described as being well known to children's social care, either as child protection cases ('stable at this point' families) or child in need cases ('on the edge' families; see table 1). It was often unclear as to whether 'stable at this point' families had *current* contact with child protection services and this was not probed by the interviewer. It was not clear whether the 'was it, wasn't it?' or 'straightforward' cases were known to children's social care prior to the referral made by the participant. See table 1 for a detailed summary of all four types of family narrative.

### Why *these* families?

We asked the GPs to discuss the cases in which they had been personally involved. The reasons that GPs gave for

**Table 1** Whom (typology of narratives about families)?

| 'Stable at this point in time but it's a never ending story' | 'On the edge' | 'Was it, wasn't it?' | 'Fairly straightforward' |
|---|---|---|---|
| Most common narrative, N=16* | Second most common narrative, N=12* | Third most common narrative, N=9* | Least common narrative, N=3* |
| ▸ Very serious and long-term parent drug/alcohol use, mental health problems and domestic violence | ▸ Lack of boundaries for children; poor school attendance, missed medical appointments, concerns about nutrition and clothing | ▸ Concerns focused on possible physical or sexual abuse | ▸ These narratives were characterised by concerns about maltreatment described as 'obvious' or 'barn door' with a high level of suspicion from participants and decisive referrals to CSC or secondary healthcare |
| ▸ Extensive contact with CSC child protection services, police and drugs and alcohol services | ▸ Families suffered from: unemployment; inadequate housing; poverty; parental alcohol use or mental health problems; and overwhelming physical health and behavioural problems | ▸ Participants were very uncertain whether suspicions 'amounted to anything or not' and believed that physical or sexual abuse probably had not occurred | ▸ Narratives were characterised by participants believing that referral to social care or other agencies would result in appropriate and timely services |
| ▸ Siblings taken into care or died | ▸ Concerns about neglect and emotional abuse | ▸ They described having just enough concern to take further action | ▸ These cases were only mentioned in passing and usually as a contrast to one of the other family types, about whom participants talked in detail and at length |
| ▸ Concerns about physical neglect and emotional abuse | ▸ Accounts of intermittent and inadequate involvement from child protection services | ▸ In the context of this low level of concern, GPs described CSC response as unnecessarily heavy handed and punitive | |
| ▸ GPs believed that circumstances had recently improved for the children and felt hopeful about capacity to parent in the future | ▸ Children described as 'vulnerable' and often as currently involved with CSC as a child in need | ▸ After varying amounts of time (a few days to a year), participants reached the decision, usually in conjunction with CSC, that the child was *not* likely to have been physically or sexually abused. In the four cases of injured children, participants described on-going concerns about parental supervision (i.e.,neglect) | |
| ▸ But new stability was seen as fragile and optimism about future was cautious and uneasy | ▸ Problems experienced by GPs as overwhelming and frustrating | | |
| ▸ Perceived need for continued vigilance to spot relapses (further neglect/emotional abuse) and prevent poor child outcomes | ▸ Worry about families 'tipping over the edge' at any moment | | |

It is important to remember that these typologies of families only tell us about GP perspectives and understandings and cannot be relied on as accurate data about families.
*More narratives than families because some families had more than family classification as participant's views of the family evolved over time.
CSC, children's social care; GP, general practitioner.

choosing a particular case were: it was particularly 'challenging' or 'complex'; it was typical; it demanded a lot of time and energy or it was fresh in their mind following a recent contact with the family.

Analysis of the narratives in their entirety revealed a clear divide between 'fairly straightforward' narratives in which GPs described onward referral of concerns without further involvement and the other types of families where participants described taking responsibility and having on-going involvement with maltreatment-related concerns. There were three characteristics typical of accounts of intense or long-term involvement with maltreatment-related concerns. First, GP involvement could be justified when GPs perceived high medical need in family members, were in regular contact with the families for this reason and conceptualised their own professional response as a 'medical' one. This containment of safeguarding within a medical sphere seemed most compatible, with chaotic, neglectful families seen to be suffering a host of medical and social problems. Second, GPs appeared more motivated to intervene when the parents were perceived as 'incompetent' rather than malicious. This perspective also seemed most compatible with chaotic, neglectful families in which parents were perceived to have had a poor childhood and were struggling with a multitude of other problems. Third, GPs seemed likely to take responsibility for maltreatment-related concerns when they distrusted the contribution from children's social care. GPs distrusted input from children's social care when they

perceived this agency to be underestimating the seriousness of the problem ('on the edge' families) or to be responding in an unnecessarily aggressive and punitive manner ('was it, wasn't it' families; see table 1).

## Actions

There were seven actions that the GPs described taking in response to maltreatment-related concerns:
1. Monitoring concerns
2. Advocating for families
3. Coaching parents
4. Providing opportune healthcare for children
5. Referral to other services
6. Working with other services
7. Recording the concerns

The definitions and descriptions of each of these seven actions are given in table 2. Some of the actions were orientated towards whole families (monitoring and advocating), some towards the parents (coaching), some towards the children (opportune healthcare) and some towards other agencies (referral to and working with other services). As table 2 summarises, the GPs were very aware that their management of maltreatment-related concerns relied on regular contact with families for non-maltreatment related reasons (monitoring and opportune healthcare), help-seeking behaviour and honest disclosure of problems from adult family members (monitoring and advocating), parental engagement with general practice (coaching and advocating) and being able to offer services that parents wanted (monitoring and opportune healthcare).

Referrals to other services and joint working across services were discussed almost exclusively in relation to children's social care and paediatric services. GPs acknowledged their reliance on health visitors and GP colleagues for gathering further information (for monitoring) and, in the case of concerns about neglect, deciding whether or not to make referrals to children's social care. GPs told how they directly referred concerns about sexual or physical abuse to children's social care without consulting other primary care colleagues (tables 1 and 2). GPs were conscious that they relied on regular meetings of the primary healthcare team in order to gather wider information about families from health visitors. Health visitors were also seen as a conduit for information about children's social care input with families. For cases perceived to be urgent, health visitors were accessed via telephone or in 'corridor conversations', which were perceived to be few and far between following relocation of health visitors away from general practice.

## Facilitators and barriers
### Relationship between GPs and families

Participants described how they went out of their way and invested a significant time and effort to develop trust with parents as part of their response to maltreatment-related concerns. This was the strongest and most persistent theme across the interviews. GPs described how they cultivated a position as trusted ally— a dependable professional who had a family's best interests at heart (box 1, quote 1). Trust and engagement were seen as necessary for monitoring maltreatment-related concerns (encouraging patients to 'come through the door', seek help with parenting and honestly disclose information) and providing coaching and advocacy (encouraging parents to be receptive to advice; box 1, quotes 2 and 3 and table 2). Keeping the parents in contact with and engaged with general practice was a key motivator for the participants (box 1, quotes 4 and 5). GPs saw that it was easiest to develop trust and encourage engagement when they had something to offer the family, such as being able to meet high health need or write a letter in support of state benefits and/or housing (box 1, quotes 6 and 7). Developing trust with parents was perceived to have potential harms as well as benefits. Several participants highlighted the potential for the child's needs to be overlooked or the extent of the maltreatment 'missed' due to a focus on parental needs and the primacy of the GP-parent relationship. The GPs described themselves as consciously navigating a course between benefits and harms (box 1, quotes 8 and 9).

### Relationship between GPs and health visitors

In all but three interviews, GPs revealed dependence on health visitors in their responses to maltreatment-related concerns and talked about this professional group far more than any other. The access to health visitor knowledge, assessments and time was seen as a necessarily facilitator of monitoring, referral to children's social care and working with children's social care (table 2). However, the two health visitors in our sample did not see GPs as central to their safeguarding work unless there was a 'medical' element to the concern (box 2, quotes 1 and 2). The two health visitors believed GPs had much more limited knowledge than they did (box 2, quote 3) and were ignorant of important information, despite having regular contact with these families (box 2, quotes 4 and 5). The health visitors viewed GPs as keen to avoid or off-load child protection work (box 2, quotes 5 and 6). Health visitors and GPs recognised that their relationship was undermined by the trend towards relocation of health visitors away from general practice (box 2, quotes 7 and 8). The responses that GPs described as reliant on health visitor input and communication should be viewed in the context of the probably imperfect and unequal relationship between the two professionals.

### Relationships between GPs and other professionals

In comparison to their description of working with health visitors, GPs gave relatively little detail about how relationships with other professionals helped or hindered their responses. GPs wished to be seen as separate from children's social care and paediatric services, which they thought patients saw as punitive and policing (box 3, quotes 1–3). Both services were perceived to be

**Table 2** Actions

| What | For whom | How | Why | Context |
|---|---|---|---|---|
| Monitoring: keeping a 'watchful eye' on families and being 'a bit more vigilant' | Frequently 'stable at this point'. Occasionally 'on the edge' families | ▶ Using routine health checks in children and regular consultations for health problems in parents to assess well-being of children and coping/risk factors in parents<br>▶ Receiving information about family life and parenting from other family members during consultations, especially grandmothers<br>▶ Assessing the family and risk during (routine) GP postnatal home visits<br>▶ Checking the electronic health records for subsequent presentations to colleagues<br>▶ Interpreting missed appointments as a possible sign of escalating problems in the family. Usually this relied on the individual practitioner but one GP was developing a practice-wide system to capture all missed primary and secondary care appointments by children aged under 16 years<br>▶ Using primary care team meetings about child safeguarding to gather wider information, anticipate stressful or important points in a family's life, such as the birth of a new baby or to gather wider information about a family. Health visitors were essential for these meetings to fulfil a monitoring function | To ascertain whether or not there was relevant information that needed to be passed onto children's social care (in the form of a referral). Missed appointments could result in a phone call from the GP and, if necessary, a letter and/or discussion in the vulnerable families meeting | When confident that the family would seek help and disclose honest information, GPs felt comfortable with the role of monitoring and risk assessment in 'stable at this point' families. Honest disclosure and help-seeking behaviour in families relied on GPs being seen as a trusted ally.<br>Some GPs and the health visitors recognised that GP monitoring was limited due to 'health' focus without wider information. GPs relied heavily on health visitors to fulfil their monitoring role |
| Advocating: 'you've got to stand up and shout for people' (making a case to other agencies on the participant's behalf) | Frequently 'on the edge' and 'was it, wasn't it? ' families Occasionally 'stable at this point' families | ▶ Supporting requests for improved housing or benefits<br>▶ For 'on the edge' families, interceding with children's social care to make this agency recognise the seriousness of the family's problems and offer (what the GPs perceived to be) a more appropriate level of service (usually child protection services)<br>▶ For 'was it, wasn't it' families, interceding with children's social care to reduce an unnecessarily heavy handed or insensitive approach and encouraging these families to demonstrate cooperation with children's social care | Improving quality of life (housing, poverty) was perceived as directly impacting on parenting and, by this route, on child welfare<br>GPs saw many 'on the edge' children as in need of protection (and sometimes removal) in order to mitigate poor child outcomes<br>By encouraging compliance, GPs aimed to avoid things 'getting worse' for these families with an even more coercive approach from children's social care and, instead, to help the family access supportive children's social care services | The need to intercede with children's social care was seen as greatest in the 'on the edge' families whose children has suffered 'terrible neglect' over years but where maltreatment did not pose an immediate threat to child's physical safety and/or was not as 'barn door' as some of the other types of abuse |

Continued

**Table 2** Continued

| What | For whom | How | Why | Context |
|---|---|---|---|---|
| Coaching: activating of parents by attempting to shift mind-set, take responsibility for their problems and, eventually, change behaviours | Frequently 'on the edge' families | ▶ Talking to parents, usually the mother, to encourage them to 'look at different ways of thinking about things', such as realising 'that there was actually a problem with the children' or that 'stopping drinking was a good thing'<br>▶ Talking to parents, usually the mother, to encourage them to 'change their life' or 'change her behaviours' | A parent's willingness or ability to recognise that there was a problem seemed to make the difference between situation perceived as hopeful and one perceived as hopeless for the family. Parental (maternal) recognition of the problem was seen as the first step in intervening to improve the situation for the children | This was described as a difficult task that was often attempted but infrequently achieved<br>In order to have a hope of changing parental mind-set (and eventually behaviour), GPs saw that the parents needed to be engaged with primary care and to see the GP as a trusted ally<br>Coaching was facilitated by being able to offer something that the family wanted (leverage) such as letters to support benefits claims and easy access to a willing health visitor |
| Opportune healthcare: providing (missed) routine and preventive healthcare for children during consultations for other reasons | Frequently 'on the edge' families | ▶ Meeting preventive healthcare needs of the children during parent/child consultations for other reasons (eg, overdue immunisations or developmental checks)<br>▶ This had to be carried out immediately as the parents could not be relied on to come back at a later date | | |
| Referral to other services<br>Although there were mentions of referral to the police or to specialist child protection assessment clinics, these were rare. In contrast referral to children's social care and/or paediatric services were common | Frequently 'fairly straightforward' and 'was it, wasn't it' families.<br>Occasionally 'stable at the moment' families | Children's social care<br>▶ Immediately, decisively and directly following consultation with a child or parent<br>▶ After using health visitor opinion or follow-up to confirm or counter GP concerns, sometimes via an additional filter of the safeguarding lead in the practice | | Direct referrals to children's social care involved certainty about physical abuse. For emotional abuse, neglect or highly uncertain physical abuse GPs used follow-up by health visitors to scale concerns up and meet thresholds for referral to children's social care or provide reassurance and decide against referral |
| | 'Was it, wasn't it' families | Paediatric services<br>▶ Referral to hospital paediatricians for an assessment of injuries or symptoms which might be related to physical or sexual abuse<br>▶ Children referred to paediatric services were also simultaneously referred to children's social care by the GP | GPs sought a full assessment and documentation of child injuries or symptoms, including probable cause | GPs recounted stories of how paediatrician behaviour could be insensitive to GP–family relationships and did not support or encourage future referrals |

GP, general practitioner.

## Box 1 Relationship between GP and family: quotations

1. "Well, I just wanted her [the mother] to know […] there was someone steady and with their hand on the tiller." (*Participant 8; discussing an 8 year old child*)
2. "It's [the reason to develop trust] not frightening them away because, as well, there is that kind of unseen agreement between you. She is thinking: 'if this gets a bit much for me, I might be asking you for a bit more help'. 'How will you be when I ask you for more help?' and I am thinking 'if this gets too much for you I might ask you if you need more help. I want you to be accepting of that help and not worried about it." (*Participant 0, discussing a 4 year old child with older siblings*)
3. "I have no teeth to then in any way punish her [the mother] or hold her otherwise to account. All I can say is I'm disappointed that you haven't done this. […] Doctors don't go about punishing patients by and large. We rely on our encouragement and then a sort of heavy sigh and well…" (*Participant 4, discussing a 2.5 year old child*)
4. "The way general practice is set up is, is that we respond to people who decide that they want our help. […] You know what's come to you, but you don't know what's out there that isn't coming to you, that isn't choosing to come through the door, for whatever reason." (*Participant 7, discussing siblings aged 6 and 10 years old*)
5. "[If we don't engage her] that girl will shut herself and we will not be able to get all the story from her what's happening" (*Participant 15, discussing siblings aged 9 and 11 years old*)
6. "…making sure they have got the right meds, making sure that you hurry along the referrals, making sure that they are dealt with politely…." (*Participant 0, discussing a 4 year old child with two older siblings*)
7. "because we can actually give them what they think they want but there may be a trade-off. 'I can get what I want, if I accept this.'"(*Participant 0, discussing a 4 year old child with two older siblings*)
8. "So I was kind of...I'm try...I'm trying to steer a line between, um, keeping her [the mother] informed and feeling I'm kind of...and not wanting to miss anything." (*Participant 8, discussing an 8 year old child*)
9. "So it's a fine balance to make and sometimes as a professional you have to make sure everybody is safe and at the same time you keep that confidence." (*Participant 15, discussing siblings aged 9 and 11 years old*)

*All quotations in this box are from GP participants.*
GP, general practitioner.

## Box 2 Relationship between GPs and health visitors: quotations

1. Interviewer: "And how do you see, how does a GP or that GP surgery support you with what you're doing with the family?" Respondent: "I don't know, yeah. I, I, I mean I'll ring up and I'll say I'm worried and they'll, but yeah, I don't know really." (*Participant 2, discussing siblings aged 2 and 3 years old*)
2. "Unless it was a health need as in, did I see a burn on the arm, then I might [inform the GP]. But certainly if it was just emotional kind of neglect or anything like that, I wouldn't routinely phone the GP there and then to say I'd made the referral." (*Participant 16, talking generally*)
3. "Certainly in my experience I've never been informed of anything that I didn't know of via a GP." (*Participant 16, talking generally*)
4. "I don't think they were aware, and certainly weren't aware that she was going off on drinking binges and leaving the children." (*Participant 16, discussing siblings aged 3 and 7 years old*)
5. "I don't think they're aware of the problems" (*Participant 1, discussing four siblings under 6 years old*)
6. "…but it is worrying and it happens more often than what I think we know, that GPs avoid addressing issues." (*Participant 16, discussing siblings aged less than 1 and 2 years old*)
7. "I think they're, again, a family that probably take up quite a lot of the GP's time so the GP's quite happy to sort of share it out." (*Participant 1, discussing four siblings under 6 years old*)
8. "I think ultimately being based in the same building, seeing people day to day, you know in the kitchen, putting the kettle on, that kind of daft thing does build a good relationship" (*Participant 16, discussing siblings aged 3 and 7 years old*)

*All quotations in this box are from the two health visitor participants.*
GP, general practitioner.

### 'A very medical role'

Just as the two health visitors confined the GPs role to a 'medical' one, so the GPs in the sample framed their responses as 'medical'. Framing of responses and problems as 'medical' was one way that the GPs justified and legitimised their on-going involvement with families who had known maltreatment-related problems. In this way, the medicalisation of maltreatment-related concerns and responses acted as a facilitator of GP action.

On-going involvement with the maltreatment-related concerns was justified first and foremost in terms of high medical need in the families (box 4, quote 1). Several GPs stated or implied that contact with families for maltreatment-related concerns in the absence of 'medical' need was not a legitimate part of the GP's role (box 4, quote 2). The theoretical distinction between 'medical' and 'social' problems was used by participants to delineate where the GP could legitimately be involved with maltreatment-related concerns. However, elsewhere in the interviews, this neat distinction was challenged. 'On the edge' families were described as presenting indiscriminately with health and social welfare need (box 4, quote 3) and one participant described how the complex mix of family need forced her to step into

insensitive to the GP's position: children's social care did not provide necessary feedback to the GP (box 3, quote 4) and paediatric services could unthinkingly and unnecessarily damage hard-earned GP–patient relationships (box 3, quote 5). The one-way flow of information share with children's social care was seen to be exacerbated by lack of personal relationships between GPs and social workers and high staff turnover within children's social care. In the case of paediatric services, GPs were able to draw on personal contacts to deliberately seek out trusted paediatricians (box 3, quote 6).

**Box 3 GPs and other professionals: quotations**

1. "I think a lot of people view social services as their only job is to take children away." (Participant 13, discussing unborn child)
2. "she [the paediatrician] is seen as just there to check up on you." (Participant 0, discussing 13 months old child)
3. "that can affect your relationship with the patient because then they lump you with social services and see you as part of the people trying to take away their child." (Participant 13, discussing unborn child)
4. "You don't get information from social services. They don't let you know, unless there happens to be a reason for them ringing because they want information from us." (Participant 7, discussing unborn child)
5. "They saw a general paediatrician, he just thought it was rough play and he didn't see why on earth I'd sent them along, which completely undermined our position. The last thing we needed was to get a secondary care response that did that because it then became more difficult to engage them at a child in need level because it's much more voluntary, isn't it?" (Participant 5, discussing three siblings aged between 5 months and 3 years old)
6. "So I think that would—that's—I think it's very important that as clinicians we sit and talk to each other about who we trust and who we don't trust in secondary care as well." (Participant 2, talking generally)

*All quotations in this box are from GP participants.*
GP, general practitioner.

**Box 4 GPs and other professionals: quotations**

1. Interviewer: "And what do you think is your role as a GP for them?" Respondent: "Well, I...I...I think that we'll always have a very medical role for this family. They're very...they have very great medical needs so they...that's kind of...although it's difficult, is the relatively easy bit. I mean, how we tap into the sort of welfare issues of families and children, I think is, um, much more difficult, much more difficult." (Participant 5, discussing 4 years old with four siblings)
2. "...arranging follow up for the purposes of reviewing concerns around umm, safeguarding, I wouldn't see as part of our role." (Participant 7, discussing siblings aged 6 and 10 years)
3. "They used to come for their medications. They used to come for all these letters for Social Services, letters for something, housing, benefit or something or something." (Participant 15, discussing 2 years old)
4. "...maybe we should just be saying, well, I'm sorry, but there's nothing I can do or, you know, I am the GP, I'm not the social worker. If she's not going to school, you know, you'll have to phone social services or somebody else who can do this, because that's not my job. And maybe we sort of just blurred boundaries too much by taking on work that possibly isn't really appropriate for us to do." (Participant 10, discussing three siblings aged between 9 and 16 years old)

*All quotations in this box are from GP participants.*
GP, general practitioner.

multiple roles, some of which were perceived to be contested (box 4, quote 4). The extent and nature of the GP role was a difficult and slippery concept for the GP participants.

Figure 1 summarises the relationship between the families that GPs described responding to, the actions they described taking and the important barriers and facilitators that helped or hindered these responses.

## DISCUSSION
### Summary of findings
GPs described being actively involved with the management of (possible) child neglect and emotional abuse and much of their response was aimed at the parents or the whole family. GPs described seven important responses: monitoring, advocating, coaching, providing opportune healthcare, referring to other services, working with other services and recording. Three main facilitators and barriers emerged from the data. First, help-seeking behaviour and honest disclosure from parents were deliberately encouraged by the GPs who described a significant effort in establishing a trusting and reciprocal relationship. Parental engagement with general practice and help-seeking behaviour were seen as necessary for GP responses to have any chance of changing parental mindset or behaviour, and thereby improving circumstances for the child. Second, information and support from health visitors were threatened by

mismatched expectations and relocation of health visitors. Third, conceptualisation of the problem and the response as 'medical' permitted and justified GP involvement. GPs saw some limitations of the way that they responded including: working within a reactive system, potentially prioritising the needs of the parent over those of the child or 'missing' things.

This study describes responses that are feasible where there is some expertise and interest within general practice. Despite our case-based approach and although accounts were detailed, candid and included emotion and uncertainty, it is possible that some GPs recounted what they thought they *should have done* rather than what they actually did. This study was not designed to quantify how far the family types represent maltreatment-related concerns among all GPs in England but the families described by our participants are likely to be familiar within general practice. Descriptions of 'on the edge' and 'stable at this point families' were compatible with other descriptions of families and adults with social welfare problems in this setting.[26] 'On the edge' narratives resonated with another well-known presentation: the 'heart-sink' patient. 'Heart-sink' patients have been described as those whose chronic and multiple problems cannot be cured or solved and which evoke exasperation, defeat and helplessness in the GP.[27 28]

Equally, although we do not know how far the seven responses are being used in general practice more widely, they do reflect core GP skills. Monitoring, which can also been termed review or 'watchful waiting' is a

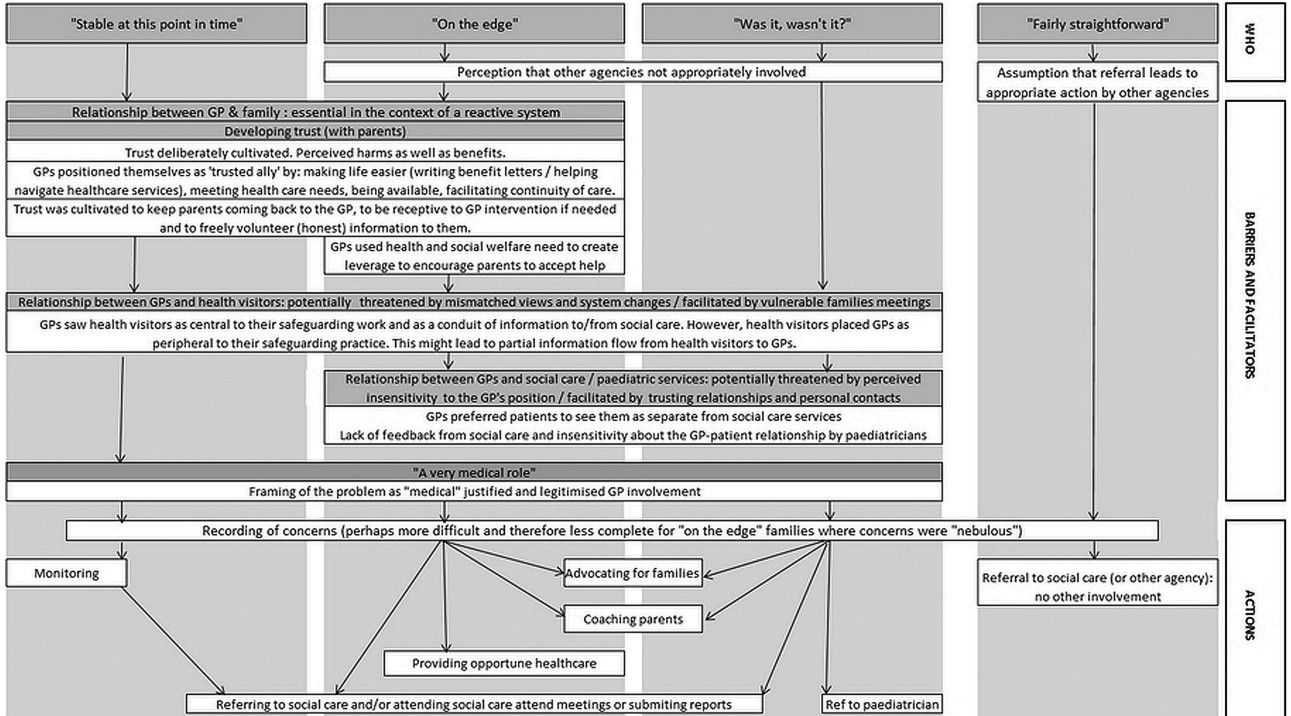

**Figure 1** To whom were general practitioners responding to, what actions did they take and what were the facilitators and barriers of these actions?

substantial part of GP practice and has been used as part of proactive management for other groups who present with a mixture of social and welfare problems, such as the frail elderly.[29] Acting as an advocate to help patients access and navigate services within and beyond the National Health Service constitutes part of managing chronic health conditions in general practice and is expected by its patients.[30–33]

Coaching incorporates elements common to promoting 'self-management' of chronic disease and 'motivational interviewing', in which professionals attempt to activate a response from patients by encouraging them to take responsibility for their own health.[34] Providing opportune healthcare as a routine part of consultations has been long considered a fundamental part of the GP consultation.[35] Feedback from participants on provisional results supported the interpretation of monitoring, advocating, coaching and opportune healthcare as core GP work. Several GPs stated that they would use these skills more widely, specifically for patients with cancer or multimorbidities.

In summary, responses to maltreatment-related concerns can be located as an extension of 'normal' GP work rather than an isolated or peripheral part of their professional activity. This was explicitly recognised by some of the GPs in our sample and by some of the GPs in the mixed methods study by Tompsett et al.[20]

The findings of our study confirm those from the only other empirical study on responses to maltreatment-related concerns by GPs in England.[20] In this study, Tompsett et al outlined four roles that the GP was

perceived to play and three of them overlap substantially with findings from our study. The 'case holder' role was similar to the role that the GPs in our sample described for 'on the edge' and 'stable at this point' families. Similar to our study, the Tompsett et al's[20] study suggests that GPs might have the biggest role to play for children with chronic neglect, that GPs feel the need to keep their involvement within a 'medical' sphere, that health visitors are a key professional in GP's safeguarding responses and that building rapport with parents and providing follow-up are good practice strategies in this area. The study by Tompsett et al and other qualitative studies also report that GP responses to social welfare concerns in children, including concerns about child abuse or neglect, are often aimed at parents.[17–20] Table 3 describes how our findings confirm and extend Tompsett et al's work by (1) providing a detailed description of the monitoring, coaching, advocating and providing opportunistic preventive healthcare that were part of their 'case-holder' role and (2) by suggesting that the four roles are differentially adopted according to how family problems are understood by the GP (ie, according to family type). Our results provide a sufficiently high level of detail about GP actions and their context that they can be used as a starting point to develop relevant interventions.

The GPs in our sample saw the potential for benefit and harm in their approach to maltreatment-related concerns. Many of these overlap with the benefits and harms which have been attributed to the GP–patient relationship not only in the study by Tompsett et al

**Table 3** Comparison of our findings with study by Tompsett et al[20]

| Four roles outlined by Tompsett et al | Relevant findings from our study | |
|---|---|---|
| | Similarities | What our study adds |
| The case holder: GP has on-going relationship with family before, during and after referral to children's social care. This role builds on voluntary disclosure and establishing trust over time with the parents. This role was clearly identified by GPs but not recognised so much by the stakeholders | Comparable to the role that GPs in the sample described in relation to 'stable at this point', 'on the edge' and 'was it, wasn't it?' families, both in the on-going nature of the relationship with families and in the reliance on voluntary disclosure and trust by parents. This was the most commonly described role by the GPs in our sample | This role might be performed most commonly where<br>▶ Families had multiple health problems (including those caused by child neglect) which<br>  ▶ Provided a reason for repeated contact<br>  ▶ Legitimised GP intervention in child safeguarding concerns<br>  ▶ Offered opportunity for establishing trust and reciprocity and encourage help-seeking behaviours by meeting high need<br>▶ GPs perceived that children's social care was not/not likely to offer appropriate services<br>▶ GPs could construct concerns as due to 'incompetent' (rather than 'malicious' parenting) which allowed sympathy with the parents and facilitated on-going GP involvement<br>These factors were typical of families who prompted concerns about chronic *neglect*<br>The 'case-holder' role also included monitoring, coaching, advocating and providing opportune preventive healthcare |
| The sentinel: GP identifies child maltreatment and refers the concern to children's social care or other health services | Comparable to the role for families with 'fairly straightforward' concerns (infrequently described). Here concerns were referred onwards with no further involvement | This role might be performed most commonly where<br>▶ GPs perceived that other agencies responded (or would respond) appropriately This was typically in cases of concerns about *physical abuse* or, less frequently, an episode of acute neglect |
| The gatekeeper: GP provides information to other agencies so that those agencies can make decisions about access to services | This role was not directly comparable to any described by the GPs in the sample | The GPs did offer information to children's social care, especially for 'stable at this point' families. However, this information was unprompted and resulted from on-going monitoring and risk assessment for families with a history of very serious child-maltreatment concerns who had achieved a fragile stability |
| Multiagency team player: GP has continued engagement with other professionals outside the practice. This role is fulfilled when GP contributes actively to children's social care child protection processes | Comparable to the few instances in which GPs described working with children's social care and actively participating in their child protection processes | This role might be performed most commonly where<br>▶ GPs knew the families well and did not trust children's social care to offer appropriate services AND<br>▶ GPs perceive that there were medical issues giving them a unique medical perspective |

GP, general practitioner.

about child maltreatment but also in qualitative studies about the management of chronic conditions. A trusting and constant doctor–patient relationship has been seen by doctors and patients as facilitating honest disclosure of hardships (such as domestic violence and past abuse), to help patients cope with these issues,[36] to offer GPs a mechanism for changing patient attitudes and behaviour[34 36] and to be a way of helping the child when the

principle patient is the parent.[20] However, GPs also agree that if the relationship is not sufficiently strong, attempting to 'coach' patients might scare them away from using services[34] and a dysfunctional doctor–patient relationship might promote tolerance of 'bad' behaviour by doctors or may make GPs more likely to miss new and serious symptoms.[36 37] GPs have previously recognised that building relationships with parents may come at the cost of overlooking the child's needs.[20] Analyses of maltreatment-related child deaths suggest that therapeutic relationships can be very dangerous for the child if professionals do not recognise disguised compliance (apparent cooperation by parents to diffuse professional intervention) or if empathy with parents is accompanied by 'silo' working (failure to look at a child's needs outside of their own specific brief).[38]

The GPs in our sample described how they sought to establish a trusting relationship with the families to encourage engagement with general practice, disclosure of difficulties and acceptance of help and advice. We did not seek the views or experiences of parents and children. However, there is a considerable evidence from other qualitative studies that families perceive GPs as dismissive, unapproachable and/or judgemental,[39 40] are reluctant to confide in the GP[41] or to present[42] and perceive their relationship with the GP to be meaningless or non-existent.[20] If the families described in our sample had a similarly negative perception of the GP service, this would undermine any credible chance that the seven actions could work in the way that the GPs hoped. It is also possible that further responses might be identified in different sample of GPs.

The perspectives and experiences of parents and children are an important avenue for future research. Although there was a substantial overlap between our findings and those from the only other empirical study about GPs and wider responses to maltreatment-related concerns, it would be helpful to repeat our study in a different sample of GPs to identify any additional responses. Future studies are needed to evaluate the impact of the responses we have identified on children and families who prompt maltreatment-related concerns in general practice. Such studies should take into account the considerable skill required to use the therapeutic relationship for monitoring and coaching, the potential for more harm than good and that the responses may only be considered acceptable for concerns about neglect or emotional abuse and/or feasible for a subset of help-seeking families.

## Implications

▶ Policy and research focus should be broadened to include direct intervention by GPs for families who prompt maltreatment-related concerns, as well as GP referral to children's social care and participation in social care processes. The actions we identified provide detailed exemplars of direct intervention.

▶ A shift in thinking to incorporate core GP skills such as advocating, coaching and providing opportune healthcare into 'safeguarding' activity might make this work more central and relevant to GPs who do not consider themselves to have specialist expertise in this area. It is, however, also possible that labelling this work as 'safeguarding' might make it more difficult for GPs to respond.

▶ As the responses represent core skills and activities of general practice which are used for other patient groups, there is likely to be significant existing skill within general practice. However, it is possible that GPs more generally might not have the time or inclination to use these skills in relation to maltreatment-related concerns.

▶ Our study suggests that the GP might be a very important professional for families who present regularly to general practice with a high health need. GPs might be able to impact on child outcomes through treating health needs of the parents and building a therapeutic relationship with the parents. We do not know what proportion of families with maltreatment-related concerns fit this description.

▶ Funding is needed to develop a model of response to child maltreatment in general practice which incorporates the seven responses we identified (as well as any additional responses from future studies). Any such model must prioritise the therapeutic relationship and establish genuine help-seeking behaviour in parents, while also recognising the potential harms of this approach. The concerns about discouraging families from presenting to healthcare services should be taken seriously. This research will also be pertinent to developing the role of 'lead professional' for GPs.

▶ Models of GP practice in relation to child maltreatment must be rigorously evaluated for efficacy, safety and cost.

**Acknowledgements** The authors would like to thank the professionals who gave up their valuable time to make this study possible.

**Contributors** JW designed the study, conducted the interviews, analysed and interpreted data. She is the guarantor. RG and MB designed the study, supervised the analyses and contributed to the writing of the article. DG and JA contributed to the design of the study, interpretation of findings and contributed to the writing of the article. JW, RG and MB took responsibility for the integrity of the data and the accuracy of the data analysis.

**Funding** This study was funded by an MRC/ESRC interdisciplinary studentship awarded to JW (grant number G0800112).

**Competing interests** RG is supported by awards establishing the Farr Institute of Health Informatics Research at UCLP Partners from the MRC, in partnership with Arthritis Research UK, the British Heart Foundation, Cancer Research UK, the Economic and Social Research Council, the Engineering and Physical Sciences Research Council, the National Institute of Health Research, the National Institute for Social Care and Health Research (Welsh Assembly Government), the Chief Scientist Office (Scottish Government Health Directorates) and the Wellcome Trust (MR/K006584/1).

**Ethics approval** Ethical approval for the interviews and observations was given by the Central London 1 NHS Research Ethics Committee on 8 October 2010 (Reference 10/H0718/6).

**Provenance and peer review** Not commissioned; externally peer reviewed.

**Data sharing statement** No additional data are available.

**Open Access** This is an Open Access article distributed in accordance with the terms of the Creative Commons Attribution (CC BY 3.0) license, which permits others to distribute, remix, adapt and build upon this work, for commercial use, provided the original work is properly cited. See: http://creativecommons.org/licenses/by/3.0/

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
