## [Reviewer comments · BMJ Open]

Some articles will have been accepted based in part or entirely on reviews undertaken for other BMJ Group journals. These will be reproduced where possible.

ARTICLE DETAILS

TITLE (PROVISIONAL)	Responses to concerns about child maltreatment: a qualitative study of GPs in England
AUTHORS	Woodman, Jenny; Gilbert, Ruth; Allister, Janice; Glaser, Danya; Brandon, Marian

VERSION 1 - REVIEW

REVIEWER	Professor Hilary Tompsett Kingston University, London, UK I am a Professor of Social Work and was responsible for a research project to which direct reference is made in this article; the findings of this research are reviewed comparatively with the research findings from the original project
REVIEW RETURNED	01-Oct-2013

GENERAL COMMENTS	Title: I would suggest inserting “small” as descriptor to “qualitative study”. Qu. 2 Abstract: I suggest clarifying in the abstract that “maltreatment” should refer to child maltreatment (line 8), and ‘expert’ GP (line 24) should indicate expertise in child safeguarding/protection. Study Summary: Key Messages: bullet 2 refers to “necessary facilitators” as identified in the conclusions, but only mentions 2 of the 3 identified. Qu. 3 The design describes seeking “narratives” from GPs and practice staff. Would the research design more appropriately be described as stage analysis rather than narratives? While the accounts of the GPs et al are candid and clearly express their views of families, they do also reflect that their perspective is recorded at a particular moment in time on families with whom they may have a long relationship and where the families as in Table 1 may fit into more than one category and may shift over time into/out of completely new ones? While the research design is appropriate for the research question posed here, this is a suggestion more about the presentation of the unit of analysis. Qu. 8
--

	Some literature identified in attached notes (e.g., on thresholds) that is currently absent would be useful to refer to in the discussion. It is clear that the issue of thresholds in relation to children's social care is represented on p 10 as not recognizing the "seriousness" of the situation, rather than sharing a common view of seriousness. This could be better represented as a different view on "seriousness" and links strongly to the literature identifying the effect of high workloads and shortages of staff as significant to professional views on seriousness and thresholds. Key literature (as e.g., that used in Table 3 for comparison) needed to be discussed earlier in the article, and the synopsis of the study inserted as a footnote in Table 3 included elsewhere. Other aspects of Tompsett et al.'s work may be relevant to the discussion in this article with regard to not hearing children's views or losing focus on the child, e.g., on p 19 where it is identified that children's needs may be overlooked. Qu. 11 The analysis has identified 7 possible responses from a small set of GPs interviewed. It seems that these were identified from a small sample, and a question remains as to whether there might be more responses emerging from a larger sample. Would it be more appropriate to describe them as "at least" 7 potential responses? Qu. 12 • In the section (Strengths and Limitations, p3): a further limitation needs to be identified with regard to potential bias in the sample of participants (all known to the researchers) and potential impact on the researchers/analysis.• The small size of the sample should be acknowledged in the limitations, as well as stating that the research should be repeated with a larger sample before evaluating the response categories for potential benefits and harm. Suggestions for change to the title have been made above.• P 19 Relationship between GPs and Families: It would be helpful to the reader, at this point, to emphasize that the GP view could not be checked with the families within this study, as this has a significant bearing on whether these actions would be relevant to handling future cases. The study did not set out to find out the views of families themselves but It would be possible to include a more general comment regarding this as a limitation (and may be an area for future research.• See note on p 30: as evaluation, efficacy and cost were not considered within
--	--

	the study, this is a limitation, though the study was clearly never intended to cover these aspects. General Comments. Length: While the text outside boxes and tables was just over 4000 words, the extensive tables make this a long article, with a great deal of information in the tables. It may be appropriate to consider reducing some of the comments in the tables, and boxes? Roles of authors: While the section at the end identifies specific contributions from the co-authors, it would be helpful to be more specific throughout the rest of the article about who carried out each part of the research work, and to avoid the term "we" unless it refers to the authoring team. Titles of participants 14 GPs (4 experts, 10 others), 2 Practice Nurses and 2 Health visitors participated, but quotations from all refer to them as "participants" 1, 2, 3, etc.: could these be more explicitly identified? Knowing whether the GP comment was from an expert or all from the same practice would be really helpful.
--	---

- The reviewer also provided an annotated review which is available upon request.

REVIEWER	Lorraine Radford, Professor of Social Policy & Social Work University of Central Lancashire
REVIEW RETURNED	01-Oct-2013

GENERAL COMMENTS	This is a timely, interesting and well written qualitative analysis of GPs' reflections on their safeguarding practice. The authors acknowledge that, due to the small numbers of participants and convenience sampling methods used in recruitment, that the findings cannot be generalised nor taken to be typical of all GPs. I think that the issues raised by the paper are of such importance to GPs that the paper is very worthy of publication. However I feel that some revisions are needed to provide a better rationale for the methodology, contextualise the findings in relation to other research and acknowledge limitations for the conclusions that can be drawn. Methodology – The GPs were asked to talk about cases they had been involved with so are likely to have highlighted those that were most memorable or significant to them. The cases might have been selected because they were the most challenging, most difficult to off load on to another agency, longest lasting, most frequently seen cases in the surgery. They might have been selected because GPs felt they illustrated best the practice issues relevant to safeguarding or because these were cases they felt had been dealt with the most professionally. On the other hand the cases might have been selected on the basis of the GP's stereotypical ideas about what 'problem parenting' is and which 'types' of patient (low income,
--

	single mothers, young parents, ethnic minority, poorly housed, poorly educated etc) they think are most likely to fit into this category. Inevitably with a qualitative study of this kind the information gained is likely to be highly subjective and influenced by personal beliefs. But it is not clear whether other methods to seek information about practice experience and knowledge (such as using case study vignettes) might have been considered as a prompt to get GPs talking about their actual case experiences and might have produced more comparable information about how beliefs and attitudes could be influencing GPs' practice. The paper's lack of detail about the family circumstances beyond noting their 'chaotic nature' begs these questions. Were the 'families' mothers and children or both mothers and fathers? There is very little detail about what questions the GPs were asked in their interviews as the researchers wanted these to be led by the GPs. It would be helpful to explain why self-directed interview techniques were selected and seen to be the most appropriate. Some referencing to this method and its virtues and limitations would strengthen the section on methodology. Contextualising findings with other research – there is virtually no discussion of what has and has not already been done regards research on GPs and child protection. This makes it very hard for a reader to see the significance of the study. Not until further into the paper is the reader told that Tompsett et al is the only other comparable study. The description of the Tompsett study is hidden at the bottom of table 3. Table 3 is very confusing as there is no prior discussion of Tompsett to explain why this research suddenly appears in the table. The first column in the table refers to the 4 roles identified by Tompsett et al but below 5 seem to be listed. Limitations of the research – There are some very interesting findings in the paper about how GPs and some health visitors see their roles in child protection. A limitation of the research is that the list of actions may not be a description of actual practice but instead justifications for what GPs think they should or should not do.
--	--

VERSION 1 – AUTHOR RESPONSE

REVIEWER 1: Comments from email		
Comment Number	Comment	Response
1	I would suggest inserting "small" as descriptor to "qualitative study" in title	We agree that the larger the sample size, the more likely the study is to achieve saturation and report a complete set of themes or a full theory. However, we do not consider in-depth interviews with 17 professionals (14 GPs) to be a particularly small sample size for this type of in-depth qualitative analysis in a General Practice setting. In September 2013 BMJ Open published two qualitative articles in a

		General Practice setting which used interviews. The sample sizes were 12 (patients)¹ and 19 (GPs).² Nine GPs were interviewed in a Danish study about consultations concerning child health needs³ (including maltreatment) and the same authors only managed to interview 4 GPs in a study about GPs and child neglect.⁴ Not only is qualitative data very time-consuming to collect and analyse, but as they are 'elite interviewees' and therefore difficult to recruit, especially for hour long interviews.⁵ Hilary Tompsett's own study was unusually large with 33 in-depth interviews, of which 14 were with GPs.⁶ We collected 837 minutes of interview data from 17 participants (602 minutes from 14 GP participants), which is substantial. For this reason we have not changed the title. Instead, we have added this information about interview minutes and have emphasised the utility of repeating the study in a larger sample as the reviewer suggests below. We also state that our results cannot be generalised to General Practice as a whole.
2	I suggest clarifying in the abstract that "maltreatment" should refer to child maltreatment (line 8), and 'expert' GP (line 24) should indicate expertise in child safeguarding/protection.	We agree and have changed accordingly
3	Key Messages: bullet 2 refers to "necessary facilitators" as identified in the conclusions, but only mentions 2 of the 3 identified.	We agree and have added the third facilitator to the key messages.
4	The design describes seeking "narratives" from GPs and practice staff. Would the research design more appropriately be described as stage analysis rather than narratives? While the accounts of the GPs et al are	We agree that the participants often described a long relationship with families and that their view of families shifted over time. Sometimes this was evident in the accounts and sometimes, as the review says, we had more of a 'snap-shot' perspective. Where we could identify

	candid and clearly express their views of families, they do also reflect that their perspective is recorded at a particular moment in time on families with whom they may have a long relationship and where the families as in Table 1 may fit into more than one category and may shift over time into/out of completely new ones? While the research design is appropriate for the research question posed here, This is a suggestion more about the presentation of the unit of analysis.	changes in perspective over time in accounts, families were categorised as more than one "type", which is why there are more instances of each family type than there were families discussed (see legend to Table 1). To make this clearer, we have added a sentence to the 'To whom' results section on page 7. In this study, our primary focus was on GP responses. The types of families are important for understanding an contextualising the different types of responses that the participants described. The accounts suggest that different responses are used according to how the family is perceived at that time. This means that the unit of family 'type' is more important than the unit of each GP account in our analysis. However, we maintain that the analysis can be described as 'narrative'. The term 'narrative analysis' covers a number of approaches to data analysis, all sharing a focus on the way we make sense of the world through stories and an approach which uses the 'story' as a unit of analysis.⁷ ⁸ This is true of our analysis. For these reasons we have not changed the way we describe the analysis or present the unit of analysis. We are unsure what the reviewer means by the term "stage analysis".
5	Some literature identified in attached notes (e.g., on thresholds) that is currently absent would be useful to refer to in the discussion. It is clear that the issue of thresholds in relation to children's social care is represented on p10 as not recognizing the "seriousness" of the situation, rather than sharing a common view of seriousness. This could be better represented as a different view on "seriousness" and links strongly to the literature identifying the effect of high workloads and shortages of staff as significant to professional views on seriousness	We have added the literature on thresholds to the introduction.

	and thresholds.	
6	Key literature (as e.g., that used in Table 3 for comparison) needed to be discussed earlier in the article, and the synopsis of the study inserted as a footnote in Table 3 included elsewhere. Other aspects of Tompsett et al.'s work may be relevant to the discussion in this article with regard to not hearing children's views or losing focus on the child, e.g., on p 19 where it is identified that children's needs may be overlooked.	We have rewritten the introduction to include a summary of existing relevant literature and have included the summary of the Tompsett study here. We have cited other elements of Tompsett's findings in the discussion where we have added a paragraph stating what has already been suggested previous literature and what is new in this study.
7	The analysis has identified 7 possible responses from a small set of GPs interviewed. It seems that these were identified from a small sample, and a question remains as to whether there might be more responses emerging from a larger sample. Would it be more appropriate to describe them as "at least" 7 potential responses?	We agree that further responses might be identified in a larger or different sample. We have added this sentence to the discussion: "Our study identified seven potential responses to maltreatment-related concerns. It is possible that further responses might be identified in a larger sample of GPs"
8	In the section (Strengths and Limitations, p3): a further limitation needs to be identified with regard to potential bias in the sample of participants (all known to the researchers) and potential impact on the researchers/analysis. The small size of the sample should be acknowledged in the limitations, as well as stating that the research should be repeated with a larger sample before evaluating the response categories for potential benefits and harm.	We agree that our sample is not representative of all GPs in England. Only 4 of the 17 participants were known to researchers. We have added a sentence to the methods section to make this clear. To emphasis the non-representative nature of the sample and the limitations of sample size, we have added the following clause (in bold) to the strengths and limitations section on p.3 " Due to a small and non-random sample, results cannot be generalised to all General Practices in England." We have added the following sentence highlighting the utility of repeating a study in a different sample (we do not think our

		sample is particularly small – see response to comment 1 above): “It would be helpful for a similar study to be undertaken with a different sample in order to identify any additional responses.”
9	P 19 Relationship between GPs and Families. It would be helpful to the reader, at this point, to emphasize that the GP view could not be checked with the families within this study, as this has a significant bearing on whether these actions would be relevant to handling future cases. The study did not set out to find out the views of families themselves but it would be possible to include a more general comment regarding this as a limitation (and may be an area for future research. * See note on p 30: as evaluation, efficacy and cost were not considered within the study, this is a limitation, though the study was clearly never intended to cover these aspects.	We agree that readers need to be reminded that we did not seek perspectives from parents and children about GP relationships. We feel the best place for this is the discussion. There is considerable research suggesting that vulnerable adults and young people find it difficult to confide in GPs and do not view General Practice as a friendly or trusted service. We have added a paragraph at the end of the discussion describing this research and explaining why the lack of family perspectives is a limitation of our study and why it is important that future studies collect data on the perspectives of parents and children. We have emphasised in the main messages and the discussion that future research needs to evaluate efficacy, safety and cost. We feel this is sufficient to remind the reader that this study did not aim to evaluate the responses we identified.
10	While the text outside boxes and tables was just over 4000 words, the extensive tables make this a long article, with a great deal of information in the tables. It may be appropriate to consider reducing some of the comments in the tables, and boxes?	We agree that this is a long article, which contains much data. However, we feel that the key messages and Figure 1 provide a good summary of the complex data for readers to quickly navigate the paper. We also feel that the detail in the tables and the quotations in the boxes allows the reader essential insight and makes our results credible and trustworthy to the readers. For this reason, we have not changed the content of the boxes or tables.
11	While the section at the end identifies specific contributions from the co-authors, it would be helpful to be more specific	Although there was one main researcher collecting and analysing (as described in the author contribution section), there was constant and extensive feedback to and

	throughout the rest of the article about who carried out each part of the research work, and to avoid the term "we" unless it refers to the authoring team.	discussion with the other authors. It is convention to use "we" when writing papers and we have been very clear about who carried out each part of the work in the section at the end, as the reviewer notes. For these reasons, we are continuing to use "we" throughout the text" except in one specific paragraph highlighted by the review in comment 20 below.
12	14 GPs (4 experts, 10 others), 2 Practice Nurses and 2 Health visitors participated, but quotations from all refer to them as "participants"1, 2, 3, etc.could these be more explicitly identified? Knowing whether the GP comment was from an expert or all from the same practice would be really helpful.	We agree that more detail on the quoted participants would be helpful. However, due to information published in previous articles,⁹ any further information about participant's would risk disclosing the participant's identity. We agree that it would be interesting to go back to the data and ask whether responses by the expert GPs were different from the other GPs and explore variation between practices. However, the individual GP narratives formed a surprisingly coherent whole and there are unlikely to be any big differences. Further analyses are beyond the scope of the minor revisions to this paper requested by the editor.
REVIEWER 1: Comments from annotated PDF (see annotated PDF for position of comments in the paper)		
Abstract and key messages		
13 (p3)	Most of this would appear to be consistent with the guidance that is given by the BMA and RCGP. If the sample included several who are very familiar with or even contributed to guidance then that would not be surprising.	The monitoring role of GPs is clearly defined in BMA and RCGP guidance. However, advocating, coaching and providing opportunistic healthcare as a response to maltreatment-related concerns is not included in BMA or RCGP guidance. We have added a sentence to the discussion stating this.
14 (p3)	The advice would appear to focus on the identification of cases and does not appear to cover the inter-professional stages that are touched upon later in the paper. Since the subjects appear to be experts in	Our focus is on actions & decisions taken once a GP has identified actual/ possible/potential maltreatment. We have rewritten the conclusion of the abstract to be more precise and to include the inter-professional stages mentioned by the reviewer.

	the field then the more pressing problem of persuading other GPs to do likewise has not been addressed.	We have also included a sentence stating that exemplars of current practice, such as the seven actions described in this study, should be evaluated for feasibility in other (i.e. non-expert) General Practice settings.
15	I am uncertain what this entails ... I would have thought that every research method should be robust.	The inferior status of 'thematic' analysis that Braun and Clark discussed in their 2006 article persists. But, like them, I believe, that a robust and in-depth thematic analysis can be as insightful and skilful as other 'branded' analytical approaches.¹⁰ However, I agree that the word "robust" is unnecessary – we have removed it.
16	This might be a plausible aim but it is difficult to see how the results could be generalized without further testing against a more typical sample of GPs who might have expertise in other areas of practice.	See response to comment 8 and 15 above
17	This may be a finding but this issue has been well established as a critical factor in this field, so perhaps it has been confirmed in this study?	We have kept the wording in key messages but have added a sentence to the discussion stating that other studies have also reported that working with HVs and relationships with parents are important for safeguarding.
Intro		
18	I suggest this does not need to be in the plural.	Changed
19	Much of this level of activity has been described in policy documents dating from earlier versions of 'Working together.' The authors should clarify what is new in latest guidance.	We have rewritten this section and no longer emphasis the differences between the new and older versions of Working Together.
Methods		
20	This is another point at which the authors could, and should be more precise. Since only one	We have added a sentence it make it clear that the interviewer was the coder. See response to comment 11 above for issue of

	person appears to have been responsible for open coding (discussed later), this should be made clear at this point. One further issue should also be clarified. In order to understand the development of the open-coding it should be made clear whether the key analyst was the interviewer or not, as the experience of conducting the interviews will influence the coding process. More generally, the paper needs to be more explicit at each stage about who was responsible for which stages of analysis and validation (not just at the end). The use of the term 'we' should only be used to refer to all the authors.	"we".
21	It would be wrong to use the term 'theory' at this point, since all that has been developed is a set of descriptors and there is no indication that coding has progressed beyond that at this stage. It will also be important to understand how many of the new codes that were introduced in this way refer to a single case. These codes may have no semantic import beyond indicating that some aspect of an interview is anomalous. The same may also be true of codes that covered just two cases.	We have rewritten this sentence to make it clearer (and have removed the word "theory") "The abstract themes and understanding of relationships between them were refined by paying particular attention to data that did not fit and using reflections on these instances"
22	I find this discussion confusing. If the intention was to limit the research to four practices, then the size of the	We have removed the details about size of the convenience sample.

	convenience sample is irrelevant unless the selection was random (and even this is only relevant if a number refused to participate.)	
23	The study is focused on four GP practices, selected because each of them had a expert GP with the highest expertise. We are given four reasons for defining expertise, but there is no indication as to how 'highest' is defined and this should be clarified. Many GPs with expertise would qualify on at least two aspects, and those who have contributed to relevant policy would probably qualify under all four conditions. In order to understand the findings it is critical to know how many of the four GPs who were selected have contributed to policy development in the field. This is of particular importance in both understanding the relationship between written guidance, and in repeating this work with either a similar or different sampling method	We have added information about this in the methods (3/4 expert GPs had contributed to relevant policy). Now that there is no longer any description of the larger convenience sample from which the 4 practices were chosen (see response to comment 22), there is no need for the word “highest” and we have rewritten this section to exclude it.
24	This is a good point to make, but the number of cases in which this did occur is too small for further comment. It would be helpful to the reader to indicate this at this point ... rather than leaving the reader waiting to be told it is not relevant.	We agree – and have moved the information from the results to the methods as suggested.

25	A table is needed at this point giving the number of GPs in the practice, the number of GPs interviewed, whether health visitors were based at the practice or not, and the whether any of these health visitors were interviewed. Without this information it will be difficult for the readers to assess the potential relevance to their own GP practice.	We agree that this would be useful. However, due to information about these four and other practices that we have published elsewhere, such detail for each practice would allow readers to identify the four practices who had participated in this current study. This would, in turn, risk disclosing participant identity. For ethical reasons, the suggested table cannot be included. However, we have added this in the text in a way that is not disclosing (see first paragraph of methods).
26	Geographical spread needs to be clarified ? across England, a county, or area?	We have clarified that this is across England.
Results		
27	Although only two cases were discussed with more than one family, it should noted, at some later point, how many GPs within each practice were involved directly with each case. This is particularly important in terms of establishing relationships with the families in large practices.	Unfortunately, we did not collect this information.
28	Were these identified before the interviews, after the interviews but before the analysis, or as a result of the analysis?	The overarching questions were identified as a result of the analysis. We have clarified this in the text.
29	The use of the term category is misleading as it is clear that many narratives have been placed in more than one category. There are two ways that this can be	We have changed text to use the term 'code' instead of category. Further analysis is beyond the scope of these minor revisions.

	resolved. The authors could just re-use the term 'code', since the quotes will have been selected for some specific code. Alternatively, the coding system should be restructured to reflect the development of cases through time. This would allow a single code to refer to each period of time, increasing the number of units that are coded, and allow comments to be made regarding the progress of cases. This approach would also differentiate between the management of/response to cases before involvement of social services, as this is a key aspect where the research contributes some insight.	
30	This appears to be inconsistent with the comments in the preamble to these quotes. That suggested that many of the cases were classified in this way. Do the authors mean that only two of the narratives were classified in this way.	Apologies, we mean two family types not two families. We have changed the text.
31	It is disappointing that the current status of each of these cases with Children's Social Care (and other service providers) was not explored, as this determines whether the GP practice has to hold responsibility for a case that is identified or whether that responsibility lies with another agency.	Although we did not systematically collect information on the current status of each of these cases with children's social care, the interview data did suggest that the GPs employed the widest range of responses when they believed that they could not trust children's social care to take responsibility appropriately. We have added this information in the new section entitled "why these families" (see response to comment 46 below).

32	This is a potentially important issue that is already well recognized as the issue of thresholds. It identifies the problem of scarcity of resources faced by children's social care that is mirrored, of course, in GP practices. Suitable references to other literature should be made here.	We do not feel that the results section is the most suitable place to describe other literature. We have added this literature to the introduction.
33	Repeated references to Table 1 could be simplified for the reader without introducing confusion	Changed
34	It would be of some value to know which of these points have been raised at earlier points in the literature and which are new.	We have added a paragraph to the discussion stating how our results confirm and extend findings from previous studies
35	This section repeats entirely section 1. Case holder, and this then affects the numbering scheme.	Apologies - corrected
36	There are four different types of participants: the four experts, six other GPs, two health visitors and two practice nurses. With the current numbering of the participants it is far from easy, for example, whether the 'experts' were responsible for most of the comments and/or whether this was because their narratives were longer.	We agree that it would be interesting to analyse differences between the different types of participant and also between practices. This was not part of the analyses undertaken for this study and is beyond the minor revisions requested by the editor. However, we will consider this as further analyses if when resources allow.
37	The comments from the health visitors are too small for inclusion within this analysis except where these refer to the same cases discussed by (one or more) GPs.	We agree that the sample of HVs is very small and have been very clear about this. This results section compares the HV and GP accounts and so draws on 16 interviews, not just two. We have chosen to keep these results in the paper as they show how GPs might not be able to fulfil the role that they implicitly lay claim to. The GPs say that they need HVs to have a credible chance of monitoring families and yet the HVs say that

		they do not pass on all relevant information. Reviewer 2 highlighted these results interesting and useful (se comment 48 below)
38	Commonality on this theme is not surprising if the GPs came from 4 practices with the requirement that the theme was actively discussed across the practice.	
39	Commonality on this theme is not surprising if the GPs came from 4 practices with the requirement that the theme was actively discussed across the practice.	There was no requirement that building relationships with families was a theme that was actively discussed across the practices. I am not sure how the reviewer got this impression from the paper.
Implications		
41	This is not a valid implication. GPs may have similar expertise but GPs have expertise in a wide range of conditions and do not necessarily have the time (or inclination) to add expertise in this problem area. Even if the non-expert GPs showed a suitable level of interest in these cases, the influence of an expert discussing cases within the practice cannot be discounted.	We agree that GPs more generally might not have the time or inclination to do this type of work with families who prompt maltreatment-related concerns. However, the actions that we have identified in the study represent core skills of General Practice and that the families identified in the interviews share characteristics with other larger groups of patients familiar to the GP. We have extensively references this literature in the discussion. This means that GPs are likely to have the skills to do this kind of work even if they don't have the time or inclination. We have amended the implication to make it clear that not all GPs might have time or inclination to use their skills in this way.
42	The authors should review the wording of this statement. It is possible that this would have labelling these cases in a different way may make them harder rather than easier to treat.	Changed
43	The authors should recognize that	Sentence changed to add this point

	a larger study is likely to find more than seven responses as some codes were introduced to cover special cases.	
44	This is a valid point but not really an implication of the study, as it is not considered within the study, but could be identified as a limitation.	We maintain that this is an implication of the study as the professionals in our study identified potential harms as well as benefits of their responses. We have left this unchanged.
REVIEWER 2		
45	This is a timely, interesting and well written qualitative analysis of GPs' reflections on their safeguarding practice. The authors acknowledge that, due to the small numbers of participants and convenience sampling methods used in recruitment, that the findings cannot be generalised nor taken to be typical of all GPs. I think that the issues raised by the paper are of such importance to GPs that the paper is very worthy of publication. However I feel that some revisions are needed to provide a better rationale for the methodology, contextualise the findings in relation to other research and acknowledge limitations for the conclusions that can be drawn.	No response required
46	Methodology - The GPs were asked to talk about cases they had been involved with so are likely to have highlighted those that were most memorable or significant to them. The cases might have been selected because they were the most challenging, most difficult to off load on to another agency,	We did analyse the reasons why participants discussed these families and had omitted this data to make the paper a manageable length. As Reviewer 2 has highlighted this question, we have added a new section to the results entitled "why these families?" in which we briefly outline the motivation for choosing cases. In interests of brevity, we have not included quotations and only

	longest lasting, most frequently seen cases in the surgery. They might have been selected because GPs felt they illustrated best the practice issues relevant to safeguarding or because these were cases they felt had been dealt with the most professionally. On the other hand the cases might have been selected on the basis of the GP's stereotypical ideas about what 'problem parenting' is and which 'types' of patient (low income, single mothers, young parents, ethnic minority, poorly housed, poorly educated etc) they think are most likely to fit into this category. Inevitably with a qualitative study of this kind the information gained is likely to be highly subjective and influenced by personal beliefs. But it is not clear whether other methods to seek information about practice experience and knowledge (such as using case study vignettes) might have been considered as a prompt to get GPs talking about their actual case experiences and might have produced more comparable information about how beliefs and attitudes could be influencing GPs' practice. The paper's lack of detail about the family circumstances beyond noting their 'chaotic nature' begs these questions. Were the 'families' mothers and children or both mothers and fathers? There is very little detail about what questions the GPs were asked in their interviews as the researchers wanted these to be led by the GPs. It would be helpful to explain why self-directed interview techniques were selected and seen to be the most appropriate. Some referencing to this method and its virtues and limitations would strengthen the section on	present a summary of results. We did not include more details about family circumstance in order to make the paper a manageable length. We have not added more detail for this same reason. My thesis contains a high level of detail, including about family characteristics and about the structure of the interviews. It will be published in 2014. I have added this sentence to the methods to direct readers to my full thesis should they require this extra detail. "This study was conducted as part of a PhD award and more detailed results can be found in the first author's thesis, when published." We have added a paragraph to the methods section explaining why we chose to elicit stories based on experience from participants (rather than using vignettes or using a more structured question approach). We have included references in this section.
--	---	---

	methodology.	
47	Contextualising findings with other research - there is virtually no discussion of what has and has not already been done regards research on GPs and child protection. This makes it very hard for a reader to see the significance of the study. Not until further into the paper is the reader told that Tompsett et al is the only other comparable study. The description of the Tompsett study is hidden at the bottom of table 3. Table 3 is very confusing as there is no prior discussion of Tompsett to explain why this research suddenly appears in the table. The first column in the table refers to the 4 roles identified by Tompsett >et al but below 5 seem to be listed	We have added a paragraph to the discussion explaining how our results confirm and extend previous research and have extended our reference to existing literature in the introduction. We have moved the first mention of the Tompsett study to the introduction and placed the study design there too. We have corrected the numbering problem with Table 3.
48	Limitations of the research - There are some very interesting findings in the paper about how GPs and some health visitors see their roles in child protection. A limitation of the research is that the list of actions may not be a description of actual practice but instead justifications for what GPs think they should or should not do.	We have added a sentence to the second paragraph of the Discussion to remind the reader of this limitation: it is possible that the participants told us what they thought they should have done rather than what they did.

1. Jones S, Hanchard N, Hamilton S, Rangan A. A qualitative study of patients' perceptions and priorities when living with primary frozen shoulder. *BMJ Open* 2013;3(9):e003452.
2. Roberts JH, Crosland A, Fulton J. "I think this is maybe our Achilles heel..." exploring GPs' responses to young people presenting with emotional distress in general practice: a qualitative study. *BMJ Open* 2013;3(9):e002927.
3. Lykke K, Christensen P, Reventlow S. The consultation as an interpretive dialogue about the child's health needs. *Family Practice* 2011;0:1-7.

4. Lykke K, Christensen P, Reventlow S. "This is not normal ... "--signs that make the GP question the child's well-being. *Fam Pract* 2008;25(3):146-53.
5. Green J, Thorogood N. *Qualitative methods for health research*. 2nd Edition ed: Sage Publications Ltd, 2009.
6. Tompsett H, Ashworth M, Atkins C, Bell L, Gallagher A, Morgan M, et al. *The child, the family and the GP: tensions and conflicts of interest for GPs in safeguarding children May 2006-October 2008 Final report February 2010*. London: Kingston University, 2010.
7. Liamputtong Rice P, Ezzy D. *Qualitative research methods: A health focus*. Victoria, Australia:: Oxford University Press, 1999.
8. Creswell JW. *Qualitative inquiry & research design: Choosing among five approaches*. London: Sage Publications, Inc, 2007.
9. Woodman J, Allister J, Rafi I, de Lusignan S, Belsey J, Petersen I, et al. Simple approaches to improve recording of concerns about child maltreatment in primary care records: developing a quality improvement intervention. *Br J Gen Pract* 2012;62(600):e478-e86(9).
10. Braun V, Clarke V. Using thematic analysis in psychology. *Qualitative research in psychology* 2006;3(2):77-101.

VERSION 2 – REVIEW

REVIEWER	Professor Hilary Tompsett Kingston University and St George's University of London, England, UK I am a Professor of Social Work and was responsible for a research project to which direct reference is made in this article; the findings of this research are reviewed comparatively with the research findings from the original project.
REVIEW RETURNED	25-Oct-2013

GENERAL COMMENTS	Thank you for the opportunity to re-review this article, after revisions in the light of comments previously made. Thank you also to the authors for their comprehensive and considered responses. I look forward to the publication of this highly topical article and to the debates that I hope it will generate.
--